# Assessing the Effects of Time Interpolation of NDVI Composites on Phenology Trend Estimation

Xueying Li [1,2,3], Wenquan Zhu [1,2,*], Zhiying Xie [1,2], Pei Zhan [1,2,4], Xin Huang [1,2], Lixin Sun [1,2] and Zheng Duan [3]

1. State Key Laboratory of Remote Sensing Science, Jointly Sponsored by Beijing Normal University and Aerospace Information Research Institute of Chinese Academy of Sciences, Faculty of Geographical Science, Beijing Normal University, Beijing 100875, China; xueying.li@nateko.lu.se (X.L.); xiezy@mail.bnu.edu.cn (Z.X.); peizhan@nuist.edu.cn (P.Z.); huangxin@mail.bnu.edu.cn (X.H.); 201821051190@mail.bnu.edu.cn (L.S.)
2. Beijing Engineering Research Center for Global Land Remote Sensing Products, Faculty of Geographical Science, Beijing Normal University, Beijing 100875, China
3. Department of Physical Geography and Ecosystem Science, Lund University, 22362 Lund, Sweden; zheng.duan@nateko.lu.se
4. School of Applied Meteorology, Nanjing University of Information Science & Technology, Nanjing 210044, China
* Correspondence: zhuwq75@bnu.edu.cn

**Abstract:** The accurate evaluation of shifts in vegetation phenology is essential for understanding of vegetation responses to climate change. Remote-sensing vegetation index (VI) products with multi-day scales have been widely used for phenology trend estimation. VI composites should be interpolated into a daily scale for extracting phenological metrics, which may not fully capture daily vegetation growth, and how this process affects phenology trend estimation remains unclear. In this study, we chose 120 sites over four vegetation types in the mid-high latitudes of the northern hemisphere, and then a Moderate Resolution Imaging Spectroradiometer (MODIS) MCD43A4 daily surface reflectance data was used to generate a daily normalized difference vegetation index (NDVI) dataset in addition to an 8-day and a 16-day NDVI composite datasets from 2001 to 2019. Five different time interpolation methods (piecewise logistic function, asymmetric Gaussian function, polynomial curve function, linear interpolation, and spline interpolation) and three phenology extraction methods were applied to extract data from the start of the growing season and the end of the growing season. We compared the trends estimated from daily NDVI data with those from NDVI composites among (1) different interpolation methods; (2) different vegetation types; and (3) different combinations of time interpolation methods and phenology extraction methods. We also analyzed the differences between the trends estimated from the 8-day and 16-day composite datasets. Our results indicated that none of the interpolation methods had significant effects on trend estimation over all sites, but the discrepancies caused by time interpolation could not be ignored. Among vegetation types with apparent seasonal changes such as deciduous broadleaf forest, time interpolation had significant effects on phenology trend estimation but almost had no significant effects among vegetation types with weak seasonal changes such as evergreen needleleaf forests. In addition, trends that were estimated based on the same interpolation method but different extraction methods were not consistent in showing significant (insignificant) differences, implying that the selection of extraction methods also affected trend estimation. Compared with other vegetation types, there were generally fewer discrepancies between trends estimated from the 8-day and 16-day dataset in evergreen needleleaf forest and open shrubland, which indicated that the dataset with a lower temporal resolution (16-day) can be applied. These findings could be conducive for analyzing the uncertainties of monitoring vegetation phenology changes.

**Keywords:** vegetation phenology; phenology trend; NDVI composites; time interpolation

## 1. Introduction

The vegetation phenology refers to the physiological and reproductive phenomenon of vegetation in an annual cycle, which is a robust and sensitive indicator of climate change [1–4]. Shifts in vegetation phenology can regulate interactions between vegetation and climate change by influencing the structure and functions of the terrestrial ecosystem [5–7]. The availability of accurate vegetation phenology shifts has significant implications for promoting the understanding of vegetation responses to climate change [8] and improving terrestrial ecosystem process models [9] and prediction skills in crop yield production [10].

Shifts in vegetation phenology across regional and global scales were frequently derived by using vegetation indices (VIs) from satellite remote sensing data at various spatial and temporal resolutions [11–19]. The accuracy of phenology trend estimation can be influenced by multiple variables such as geographical regions [20–22], vegetation types [23–25], and vegetation indexes [26–28] but mostly depends on the selection of remote sensing products, denoising methods, phenology extraction methods, and the different combinations of these factors [20,29]. Previous studies indicated discrepancies between phenology trends estimated by different remote sensing products [30–35]. For example, Peng et al. [36] investigated the shifts of spring green-up onset dates in six regularly updated land surface phenology products from Moderate Resolution Imaging Spectroradiometer (MODIS) and Advanced Very High Resolution Radiometer (AVHRR). Similar interannual shifts of green-up onset dates among all products only occurred in local regions while discrepancies were distributed across the contiguous United States. In Western Arctic Russia, the start of growing season (SOS) and the end of growing season (EOS) during 2000–2010 based on MODIS and SPOT-Vegetation datasets showed similar trends, but all were significantly different from the trend based on AVHRR data [37]. In addition, Zeng et al. [38] estimated the SOS trend over the northern high-latitude region and noticed that SOS continuously advanced from 2000 to 2010 by using MODIS data, but no advancing trends were shown in the AVHRR Global Inventory Modeling and Mapping Studies (GIMMS) time series. Such patterns have also been documented in Tibetan alpine grassland, where MODIS Normalized Difference Vegetation Index (NDVI) data captured the advancement of SOS throughout 2000–2014, but a delaying trend of the GIMMS NDVI estimated SOS was observed [39]. Discrepancies between phenology trends based on different denoising methods or extraction methods varied in research areas, research periods, and vegetation types [40–43]. For example, Zhu et al. [8] applied several commonly utilized vegetation phenology extraction methods on MOD09A1 (8-day) and MOD13A2 (16-day) datasets and found no significant differences between SOS or EOS trends derived from asymmetric Gaussian function, double logistic function, and the piecewise logistic function method. White et al. [20] compared 10 extraction methods for estimating the shifts in the start-of-spring dates based on the GIMMS NDVI dataset in North America; the results strongly suggested either no or very geographically limited trends towards earlier spring arrival. Wu et al. [19] applied six phenology extraction methods including the first-order, second-order, and third-order derivative; amplitude threshold; relative changing rate; and curvature change rate for deriving SOS and EOS from AVHRR. In the northern hemisphere, the SOS trends retrieved vary across methods from 1982 to 2018, while only the EOS trend estimated by the relative changing rate method was significantly advanced. In the southern hemisphere, EOS based on all methods demonstrated insignificant trends. Compared with denoising methods or extraction methods, the selection of datasets might be of a higher priority in vegetation dynamics monitoring [8,33].

Atmosphere conditions, such as cloud, dust, and other aerosols, can adversely affect the quality of satellite remote sensing VI data. To this regard, the maximum value composite method [44] was mostly used [45–49] and composited VI time-series data by retaining the maximum NDVI within a specific interval of days. Current VI composite products such as 15-day GIMMS NDVI 3 g data; MODIS 8-day (MOD09A1), 16-day (MOD13Q1), and 30-day (MOD13A3) data; and 10-day SPOT VGT S10 data were widely applied among

regional [32,50,51] and global scales [12,52]. VI composite products should be interpolated into daily scales for extracting phenological metrics (i.e., SOS and EOS). However, the process of time interpolation may not fully capture real daily vegetation growth, especially during greening and senescence stages (where the curve is changing fast) [53], which may further affect the estimation of phenology trends. Current time interpolation methods include linear interpolation [54], cubic spline interpolation [55], and curve-fitting methods. Curve fitting methods smooth the noise while fitting the curve into a continuous daily scale line, including asymmetric Gaussian function fitting [56], fast Fourier transform [20], and double logistic function fitting [57], but these methods were mostly applied as denoising measures in previous studies [20,56,58,59]. For investigating the effects of data temporal resolution on phenology extraction and trend estimation, some studies compared phenology metrics or trends derived from daily NDVI data to those derived from multi-day NDVI composites [25,27,60]. However, the different performances of multi-day NDVI composites compared with daily NDVI data are directly caused by the process of time interpolation, and how this process affects phenology trend estimation remains unclear.

In this study, we used MCD43A4 daily surface reflectance data to construct a single-year daily (reference) NDVI dataset denoised by the Savitzky–Golay filter, and then single-year 8-day and single-year 16-day NDVI composite datasets were further generated. Four typical vegetation types, five time-interpolation methods, and three phenology extraction methods were chosen for estimating phenology trends. The main goals are to comprehensively investigate the effects of time interpolation on phenology trend estimation in the mid-high latitudes of the northern hemisphere from 2001 to 2019 among (1) different interpolation methods; (2) different vegetation types; and (3) different combinations of time interpolation methods and phenology extraction methods. In addition, we also analyzed the differences between the trends estimated from the 8-day and 16-day composite data, which would provide instructions on selecting relatively coarse temporal resolution (i.e., 16 day) data for phenology dynamics monitoring, as they are easier for collecting and storing.

## 2. Data and Methods

### 2.1. Study Area and Sites

We selected the mid-latitude and high-latitude area as our study region because the vegetation seasonal changes here are evident (noticeable amplitudes in NDVI curves), rendering extracting accurate phenological metrics possible [18]. In addition, NDVI datasets here are least contaminated by solar zenith angle effects [50,61]. Four typical widely distributed vegetation types (deciduous broadleaf forest (DBF), evergreen needleleaf forest (ENF), grassland (GRA), and open shrubland (OSH)) in the mid-high latitudes of the northern hemisphere (23.5°–70°N) were chosen as the study area (Figure 1). Vegetations around long-running experiment sites are usually well protected; thus, we selected sites with long-term (at least 10 years) observations from Fluxnet (https://fluxnet.fluxdata.org/sites/site-list-and-pages/, accessed on 20 March 2020; https://ameriflux.lbl.gov/, accessed on 23 March 2020; http://www.europe-fluxdata.eu/, accessed on 23 March 2020) and Phenocam (https://phenocam.sr.unh.edu/, accessed on 5 April 2020).

First, we filtered sites with the MCD12Q1 Land Cover Type product. If the vegetation type marked at each site was the same as the type in all $3 \times 3$ pixels (1500 m $\times$ 1500 m at 500 m resolution) centered on the site location, then the site was retained; otherwise, it was removed [62]. Second, in order to eliminate the influence of bare soil, sparse vegetation, and artificial vegetation (such as crops) on VI curves, sites meeting the following criteria in all years (2001–2019) were selected for further analysis [33,50,63–65]: (1) the mean NDVI during June–September should be higher than 0.10; (2) the annual maximum NDVI should occur during July–September; (3) the mean NDVI during July–September should be 1.2 times higher than the mean NDVI during November–March; and (4) the NDVI curve has a single growth cycle annually. Finally, according to the central limit theorem [66], data statistics will be close to normally distributed if the sample size is greater than or equal

to 30. For each vegetation type, 30 sites meeting all criteria above were selected from higher to lower latitudes, which were 72 Fluxnet sites and 48 Phenocam sites in total (Figure 1).

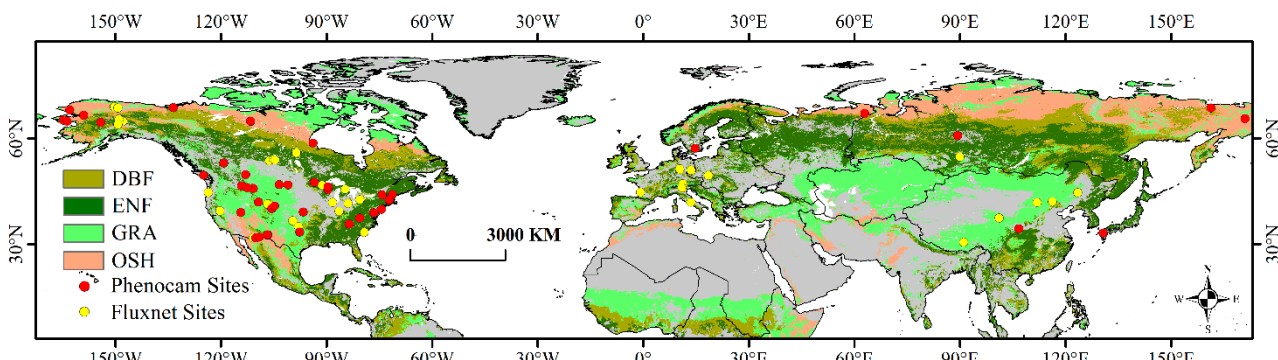

**Figure 1.** Sites and the distribution of vegetation types. DBF, ENF, GRA, and OSH are deciduous broadleaf forest, evergreen needleleaf forest, grassland, and open shrubland, respectively.

Figure 2 showed the typical NDVI curve of each vegetation type from the representative Fluxnet sites. The curve of DBF has the most apparent seasonal change (seasonal change is defined by the amplitude of the NDVI curve in Bradley et al. [67]). The curves of GRA and OSH have relatively weaker seasonal changes, and the curve of ENF has the weakest seasonal change.

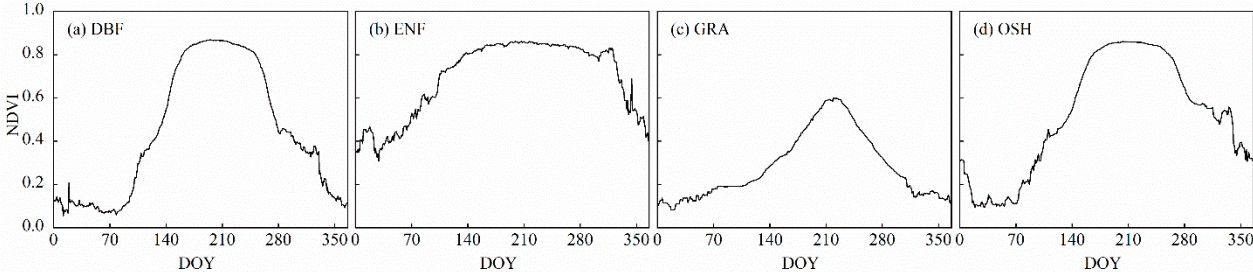

**Figure 2.** Typical NDVI curves of four vegetation types in the study area. DBF, ENF, GRA, and OSH are deciduous broadleaf forest, evergreen needleleaf forest, grassland, and open shrubland, respectively; data of (**a**–**d**) are from Fluxnet sites CA-TP3, CZ-BK1, US-Wkg, and US-IB2, respectively; all curves are denoised by the Savitzky–Golay filtering method.

### 2.2. Data and Pre-Processing

MODIS provided an 8-day surface reflectance data (MOD09A1) and a 16-day NDVI composite (MOD13A1) data with the same georeference and spatial resolution (500 m) as the MCD43A4 product. However, there exist discrepancies in data generation. In MOD09A1, the selection of pixels within the 8-day composite period is based on the minimum channel 3 (blue) value, while MOD13A1 chooses the highest NDVI value within two 8-day composite periods. In addition, the deviations appear in data gap filling. MOD13A1 uses the climate modeling grid (CMG) average vegetation index product database for gap filling, which cannot be applied to the MOD09A1 NBAR dataset. All these discrepancies may increase bias during data pre-processing. Therefore, we chose to construct the daily NDVI data and NDVI composites based on the daily surface reflectance data MCD43A4.

Firstly, we chose daily surface reflectance data for red and near-infrared ranges from MCD43A4 product with 500 m spatial resolution during 2001–2019 (https://modis.ornl. gov/globalsubset/, accessed on 20 March 2020). Surface reflectance data was then preprocessed in the following four steps to obtain single-year daily NDVI data and finally generated single-year 8-day and single-year 16-day composite data.

(1) Calculating daily NDVI during 2001–2019

Daily NDVI from 2001 to 2019 was calculated by the mean nadir BRDF (bidirectional reflectance distribution function) adjusted reflectance (NBAR) values taken over $3 \times 3$ pixels in each site (Equation (1)). The days with unqualified data (labeled in "F") were skipped from the calculation:

$$\text{NDVI} = \frac{R_{NIR} - R_{Red}}{R_{NIR} + R_{Red}} \tag{1}$$

where $R_{NIR}$ is the mean of $3 \times 3$ pixels of near-infrared band surface reflectance; and $R_{Red}$ is the mean of $3 \times 3$ pixels of red band surface reflectance.

(2) Constructing single-year daily NDVI data

For the missing daily NDVI values caused by NBAR data loss among a few sites (data labeled in "F"), a linear interpolation method was applied by using NDVI values of the same day in the nearest years (before and after) to fill up. If there were no qualified NDVI values among the nearest years, the multi-year (2001–2019) mean NDVI value of this day was then used for filling up the missing value.

(3) Denoising single-year daily NDVI data

Savitzky–Golay filter [68] is a simplified least squares fit convolution that can be applied for smoothing VI curves of a set of consecutive values [58]. The filter was proved to perform well by minimizing noises (e.g., cloud-contaminated NDVI values) effectively [69,70]. It was chosen to smooth the daily NDVI data in our study (equation and parameters are shown in Table 1).

**Table 1.** Data preprocessing methods and parameter settings.

| Method | Equation | Parameter |
|---|---|---|
| Savitzky-Golay filter | $Y_j^* = \sum_{i=-m}^{i=m} C_i Y_{j+i} / N$ | $Y^*$ is the resultant NDVI value; $Y$ is the original NDVI value; $j$ is the running index of the original ordinate data; $m$ is the half-width of the smoothing window (filter); $C_i$ is the coefficient for the $i$th NDVI value of the filter; $N$ is the amount of convoluting integers; the half-width of the smoothing window is set to 1/4 of the year length (90 days); the smoothing polynomial degree is set to 4 [58]. |
| Maximum value composite | $y_{new} = MAX(y_1 + y_2 + \ldots + y_n)$ | $y_{new}$ is the resultant NDVI value; $y_n$ is the original NDVI value; $n$ is the days for compositing. |

(4) Constructing single-year composite NDVI data

The maximum value composite method [44] was chosen to generate single-year 8-day and single-year 16-day composite data (Equation and parameters are shown in Table 1).

*2.3. Methods*

2.3.1. Time Interpolation

Five commonly used functions were chosen for interpolating the 8-day and 16-day composite data (equations and parameters were shown in Table 2): (1) piecewise logistic function fitting (PL). PL fits NDVI curve to a logistic function of time with no requirements of data pre-smoothing or threshold defining [71]. The function for NDVI data with a single growth cycle is shown in Table 2. (2) asymmetric Gaussian function fitting (AG). AG based on nonlinear least squares fits to the NDVI curves [56]. It is especially suited for describing the shape of the scaled VI curves in overlapping intervals around maxima and minima. (3) polynomial curve function fitting (PCF). PCF uses the least-square regression to analyze the relationship between NDVI data and the corresponding Julian day [50]. It effectively

smooths the curve noises, and the degree of polynomial function is flexible according to the shape of the NDVI curve. (4) linear interpolation (Linear). Linear interpolation starts at the beginning of the NDVI (start point) curve and linearly constructs the missing value with the current start point and the next nearest point [54]. The function (Table 2) can also be understood as a weighted average. (5) cubic spline interpolation (Spline). Spline interpolates the data with piecewise cubic polynomials, and it allows the NDVI curve to pass through two specified endpoints with specified derivatives at each endpoint [55]. Spline is popularly used as it reduces both computational requirements and numerical instabilities arising with higher degree curves.

**Table 2.** Time interpolation methods and parameter settings.

| Method | Equation | Parameter |
|---|---|---|
| Piecewise logistic function fitting (PL) | $y(t) = \frac{c}{1+e^{a+bt}} + d$ | $y(t)$ is the resultant NDVI value at time $t$; $t$ is the Julian days; $a$ and $b$ are fitting parameters; $c$ is the amplitude of the NDVI curve; $d$ is the minimum NDVI value [71]. |
| Asymmetric Gaussian function fitting (AG) | $y(t) = w\text{NDVI} + (m\text{NDVI} - w\text{NDVI}) \times g(t)$ $(t; a_1, a_2 \cdots a_5) = \begin{cases} \exp\left[-\left(\frac{t-a_1}{a_2}\right)^{a_3}\right], & \text{if } t > a_1 \\ \exp\left[-\left(\frac{a_1-t}{a_4}\right)^{a_5}\right], & \text{if } t < a_1 \end{cases}$ | $y(t)$ is the resultant NDVI value at time $t$; $g(t)$ is the original NDVI value; $w$NDVI and $m$NDVI are the minimum and maximum NDVI value of the fitting part; $a_1$ is the position of the maximum or minimum value with respect to time $t$; $a_2$ ($a_4$) and $a_3$ ($a_5$) are the width and flatness of the right (left) half of the function [56]. |
| Polynomial curve fitting (PCF) | $y(t) = \alpha_0 + \alpha_1 \times t^1 + \alpha_2 \times t^2 + \alpha_3 \times t^3 + \cdots$ $+ \alpha_n \times t^n$ | $y(t)$ is the resultant NDVI value at time $t$; $t$ is the Julian days; $\alpha_0$-$\alpha_n$ are fitting parameters; $n$ is the degree of smoothing polynomial; the smoothing polynomial degree is set to 6 [50]. |
| Linear interpolation (Linear) | $y(t) = \frac{t-t_1}{t_0-t_1} y_0 + \frac{t-t_0}{t_1-t_0} y_1$ | $y(t)$ is the resultant NDVI value at time $t$; $t_0$ and $t_1$ are the nearest day of year (DOY) of the missing value; $y_0$ and $y_1$ are the nearest NDVI of the missing value; $t$ is the DOY of the interpolating point between $t_0$ and $t_1$. |
| Cubic spline interpolation (Spline) | $y_i(t) = a_i + b_i(t-t_i) + c_i(t-t_i)^2 + d_i(t-t_i)^3$ | $y_i(t)$ is the resultant NDVI value at time $t$ in the $i$th period; $t$ is the interpolating point between $t_i$ and $t_{i+1}$; $a$-$d$ are function parameters decided by the DOY and NDVI matrix calculation results in the $i$th period and the $(I + 1)$ th period. |

### 2.3.2. Phenology Extraction

For extracting phenological metrics, we chose three commonly used extraction methods (equations and parameters are shown in Table 3): (1) dynamic threshold (DT) method. In DT method, SOS and EOS are defined as the point in time at which the NDVI value increases and decreases to a specific level of seasonal amplitude [56]. Here, we defined the level percentage as 10%, 20%, and 30%, respectively [1,72,73]. (2) maximum rate of change (MRC) method. MRC defines the timing of the greatest NDVI change as the maximum (the left part of the curve, from the starting point to the peak) and minimum (the right part of the curve, from the peak to the ending point) values of NDVI ratio to determine the onset dates of the start and end of a growing season [50]. (3) change rate of the curvature (RCC) method. RCC defines the onset of senescence and dormancy dates as the point in time at which the rate of change in curvature in the NDVI curve exhibits local minimum or maximum values [71].

**Table 3.** Phenology extraction methods and parameter settings.

| Method | Equation | Parameter |
|---|---|---|
| Dynamic threshold (DT) | $thd = \frac{NDVI(t) - NDVI_{min}}{NDVI_{max} - NDVI_{min}}$ | NDVI($t$) is the original NDVI value at time $t$; $NDVI_{max}$ is the maximum value of the entire curve; $NDVI_{min}$ is the minimum value of the left/right curve (divided by the maximum NDVI); *thd* is the output ratio, ranging from 0–1 [56]. |
| Maximum rate of change (MRC) | $NDVI_{ratio}(t) = \frac{NDVI(t+1) - NDVI(t)}{NDVI(t)}$ | NDVI($t$) is the original NDVI value at time $t$; NDVI ($t$+1) is the original NDVI value at time $t$+1; $NDVI_{ratio}(t)$ is the NDVI ratio at time $t$ [50]. |
| Change rate of curvature (RCC) | $NDVI(t) = \frac{y(t)''}{\left(1 + y(t)'^2\right)^{3/2}}$ | NDVI($t$) is the rate of change of curve at time $t$; $y(t)'$ and $y(t)''$ are the first and the second derivative of curve at time $t$ [71]. |

### 2.3.3. Phenology Trend Estimation

Extreme values caused by weather and human interference could affect phenology trend estimation; thus, outliers of extracted phenological metrics were removed in each site based on the 30-day rule proposed by Schaber and Badeck [74]. The NDVI values were considered as outliers if the estimated residuals of the linear regression model were larger than or equal to 30 days (Equation (2)), i.e., where $|e_{ij}| \geq 30$:

$$x_{ij} = m + a_i + b_j + e_{ij} \qquad (2)$$

where $x_{ij}$ is the NDVI data of year $i$ on site $j$; $m$ is a general mean (usually set to zero for finding a well-defined solution); $a_i$ is the effect of year $i$ (2001–2019); and $b_j$ is the effect of site $j$ ($j = 1, \ldots, 120$).

Then, the trend was calculated by linear regression (Equation (3)):

$$y = ax + b \qquad (3)$$

where $y$ is SOS or EOS for 2001–2019; $x$ is the year for 2001–2019; $b$ is the intercept; and $a$ is the SOS or EOS trend for 2001–2019.

### 2.3.4. Statistical Analysis

The paired sample $t$-test was used to test if there existed statistically significant differences between each of the two experimental results (Table 4). A Kolmogorov–Smirnov (K-S) test was performed in advance to verify that all results obeyed normal distribution ($p$ values are shown in Table S1). Pairs of experimental results being tested for statistically significant differences included the following: (1) phenology trends from the daily NDVI data and NDVI composites (8-day and 16-day) among five different interpolation methods; (2) phenology trends from the daily NDVI data and NDVI composites (8-day and 16-day) among different combinations of five interpolation methods and three extraction methods (the amount of the combinations is 50 in total for each vegetation type in SOS (EOS) trend estimation); and (3) phenology trends from the 8-day NDVI composite data and 16-day NDVI composite data. The level of $p < 0.05$ indicated significant difference.

**Table 4.** Statistically significant differences between different experiment results.

| Number | Temporal Resolution | Time Interpolation Methods | Phenology Extraction Methods |
|---|---|---|---|
| (1) | 1 d vs. 8 d, 1 d vs. 16 d | PL, AG, PCF, Linear, Spline | Mean of DT, MRC, and RCC |
| (2) | 1 d vs. 8 d, 1 d vs. 16 d | PL, AG, PCF, Linear, Spline | DT, MRC, RCC |
| (3) | 8 d vs. 16 d | PL, AG, PCF, Linear, Spline | Mean of DT, MRC, and RCC |

PL, AG, PCF, Linear, and Spline are piecewise logistic function fitting, asymmetric Gaussian function fitting, polynomial curve fitting, linear interpolation, and cubic spline interpolation, respectively; DT, MRC, and RCC are dynamic threshold, maximum rate of change, and change rate of curvature, respectively; the experiment results of bold variables are tested by the paired sample $t$-test.

## 3. Results

### 3.1. Comparisons between Trends from Daily NDVI Data and NDVI Composites Based on Different Time Interpolation Methods

Trends estimated from NDVI composites were slightly different from the reference (daily) trend, but none of the interpolation methods had significant effects on trend estimation (Figure 3). The mean SOS trend of daily NDVI data was 0.07 d/year, for which its delaying rate was lower than all mean SOS trends from 8-day NDVI composite data (Figure 3a), and the SOS trends were 0.09 d/year (PL), 0.12 d/year (AG), 0.09 d/year (PCF and Linear), and 0.08 d/year (Spline), respectively. For 16-day NDVI composite data, the delaying rate of mean SOS trends based on AG (0.09 d/year) was higher than the daily SOS trend, but the SOS trends based on PL (0.04 d/year) and PCF (0.02 d/year) were lower. Linear (−0.06 d/year) and Spline (−0.08 d/year) yielded advanced mean SOS trends. For EOS trend estimation, minor differences existed between trends from NDVI composites and the daily NDVI data. The mean EOS trend of daily NDVI data was 0.03 d/year, and the advanced EOS trends were estimated based on Linear from 8-day NDVI composite data (−0.01 d/year), 16-day composite data (−0.05 d/year), and Spline from 16-day NDVI composite data (−0.01 d/year). (Figure 3b). Other trends all showed slightly delayed trends, which were 0.05 d/year (PL and AG), 0.10 d/year (PCF), and 0.01 d/year (Spline) from 8-day NDVI composite data, respectively. For 16-day NDVI composite data, the delaying rate of mean EOS trends were 0.02 d/year (PL), 0.08 d/year (AG), and 0.05 d/year (PCF), respectively.

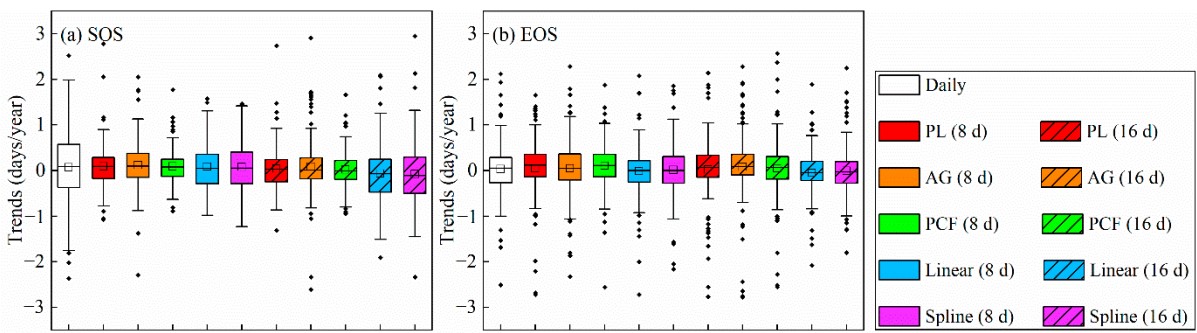

**Figure 3.** Comparisons between phenology trends from daily NDVI data and NDVI composites based on different time interpolation methods over all sites. (**a**) SOS trends comparisons, and (**b**) EOS trends comparisons. Phenology trends of daily NDVI data are unfilled, and phenology trends of NDVI composites are filled in colors; the bottom and top areas of boxes are the 25th and 75th percentiles; the lines through the boxes are the medians; the boxes designate the mean value; the diamonds beyond the ends of the whiskers are outliers; SOS and EOS are the start of growing season and the end of growing season; DBF, ENF, GRA, and OSH are deciduous broadleaf forest, evergreen needleleaf forest, grassland, and open shrubland, respectively; PL, AG, PCF, Linear, and Spline are piecewise logistic function fitting, asymmetric Gaussian function fitting, polynomial curve fitting, linear interpolation, and cubic spline interpolation, respectively.

### 3.2. Comparisons between Trends from Daily NDVI Data and NDVI Composites among Different Vegetation Types

For vegetations with apparent seasonal changes such as DBF, almost all time interpolation methods had significant effects on trend estimation (Figure 4a,e). For vegetations with weak seasonal changes such as ENF, almost no time interpolation methods had significant effects on trend estimation (Figure 4b,f). In DBF, there were significant differences between mean SOS trends estimated based on all interpolation methods and the mean SOS trend based on daily NDVI data (0.51 d/year) (Figure 4a). With the exception of the mean EOS trend (0.11 d/year) based on Spline from 8-day NDVI composite data, there were significant differences between the rest of mean EOS trends from NDVI composites and the mean EOS trend from daily NDVI data (0.40 d/year) (Figure 4e). In ENF, there was significant difference only between the mean SOS trend based on AG from 8-day NDVI

composite data (0.30 d/year) and the mean SOS trend from daily NDVI data (0.04 d/year) (Figure 4b). In GRA, the mean EOS trends based on AG from 8-day NDVI composite data (0.29 d/year) and 16-day NDVI composite data (0.25 d/year) were all significantly different from the mean EOS trend from daily NDVI data (−0.07 d/year) (Figure 4g). In OSH, the mean SOS trends based on AG, PCF, and Linear from 8-day NDVI composite data were 0.05 d/year, −0.03 d/year, and −0.04 d/year, respectively; the SOS trends based on AG from 16-day NDVI composite data were 0.18 d/year, which were all significantly different from the mean trend from daily NDVI data (−0.17 d/year) (Figure 4d). The EOS trend based on PL from 8-day NDVI composite data (0.22 d/year) was significantly different from the mean EOS trend from daily NDVI data (−0.05 d/year) (Figure 4h).

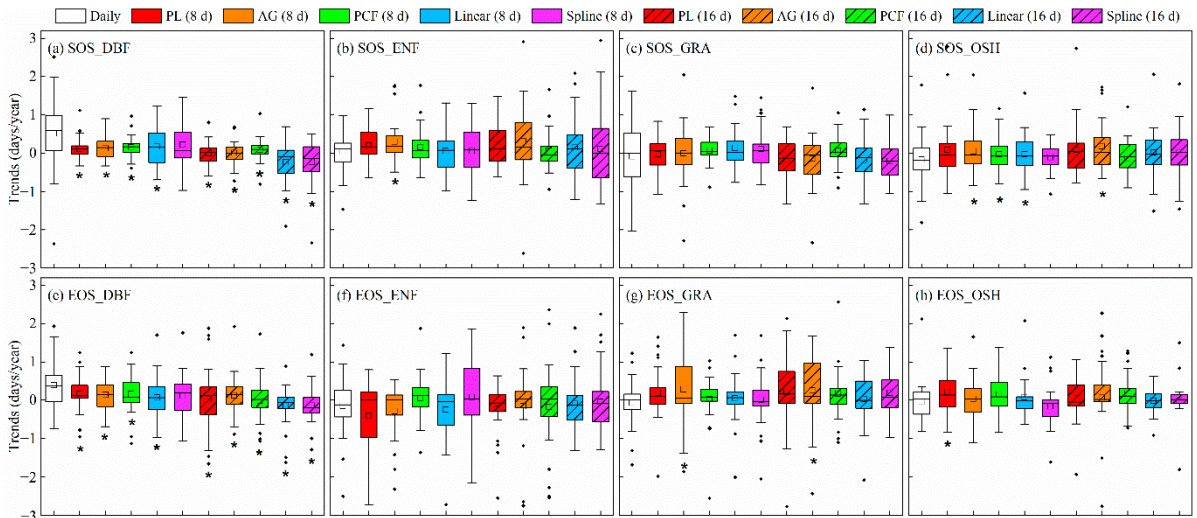

**Figure 4.** Comparisons between phenology trends from daily NDVI data and NDVI composites based on different time interpolation methods among different vegetation types. (**a**) SOS trends comparisons in DBF, (**b**) SOS trends comparisons in ENF, (**c**) SOS trends comparisons in GRA, (**d**) SOS trends comparisons in OSH, (**e**) EOS trends comparisons in DBF, (**f**) EOS trends comparisons in ENF, (**g**) EOS trends comparisons in GRA, and (**h**) EOS trends comparisons in OSH. Phenology trends of daily NDVI data are unfilled, and phenology trends of NDVI composites are filled in colors; the bottom and top areas of boxes are the 25th and 75th percentiles; the lines through the boxes are the medians; the boxes designate the mean value; the diamonds beyond the ends of the whiskers are outliers; SOS and EOS are the start of growing season and the end of growing season; DBF, ENF, GRA, and OSH are deciduous broadleaf forest, evergreen needleleaf forest, grassland, and open shrubland, respectively; PL, AG, PCF, Linear, and Spline are piecewise logistic function fitting, asymmetric Gaussian function fitting, polynomial curve fitting, linear interpolation, and cubic spline interpolation, respectively; * below the box indicates that there is significant difference ($p < 0.05$) between the phenology trend of the daily NDVI data and the trend estimated based on this time interpolation method.

### 3.3. Comparisons between Trends from Daily NDVI Data and NDVI Composites Based on Different Combinations of Time Interpolation Methods and Phenology Extraction Methods

There were 50 combinations of interpolation methods and extraction methods for each vegetation type in SOS (EOS) trend estimation, and the number of combinations for which its trends had significant differences from the trend of daily NDVI data is the largest in DBF among all vegetation types (Figure 5). In DBF, there were significant differences between SOS trends from 35 combinations and the daily SOS trend, among 30 of which included extraction methods of DT 10%, DT 20%, and DT 30% (Figure 5a). Significant differences were found between the EOS trends from 28 combinations and the daily EOS trend (Figure 5e). For 16-day NDVI composite data, significant differences were shown between the daily EOS trend and EOS trends from the combinations of PCF, Linear, Spline, and all extraction methods but only from the combinations of PCF, Linear, Spline and DT 10%, DT 20%, and DT 30% for 8-day NDVI composite data. In ENF, SOS trends from three combinations showed significant differences compared with the daily SOS trend,

which shared DT 10% as the only extraction method. (Figure 5b). EOS trends from two combinations of Spline and MRC, AG, and RCC based on 8-day NDVI composite data were significantly different from the daily EOS trend (Figure 5f). In GRA, the SOS trend from only one combination was found to have significant difference from the daily SOS trend (Figure 5c), which was PCF and DT 10% based on 8-day NDVI composite data. Moreover, there were three EOS trends from only three combinations that were significantly different from the daily EOS trend, which were all based on 16-day NDVI composite data (Spline and DT 20%, PL and DT 10%, and AG and MRC, respectively) (Figure 5g). In OSH, SOS trends from 10 combinations showed significant differences compared with the daily SOS trend (Figure 5d), while EOS trends from seven combinations were significantly different from the daily EOS trend (Figure 5h), and no combinations included DT 10%.

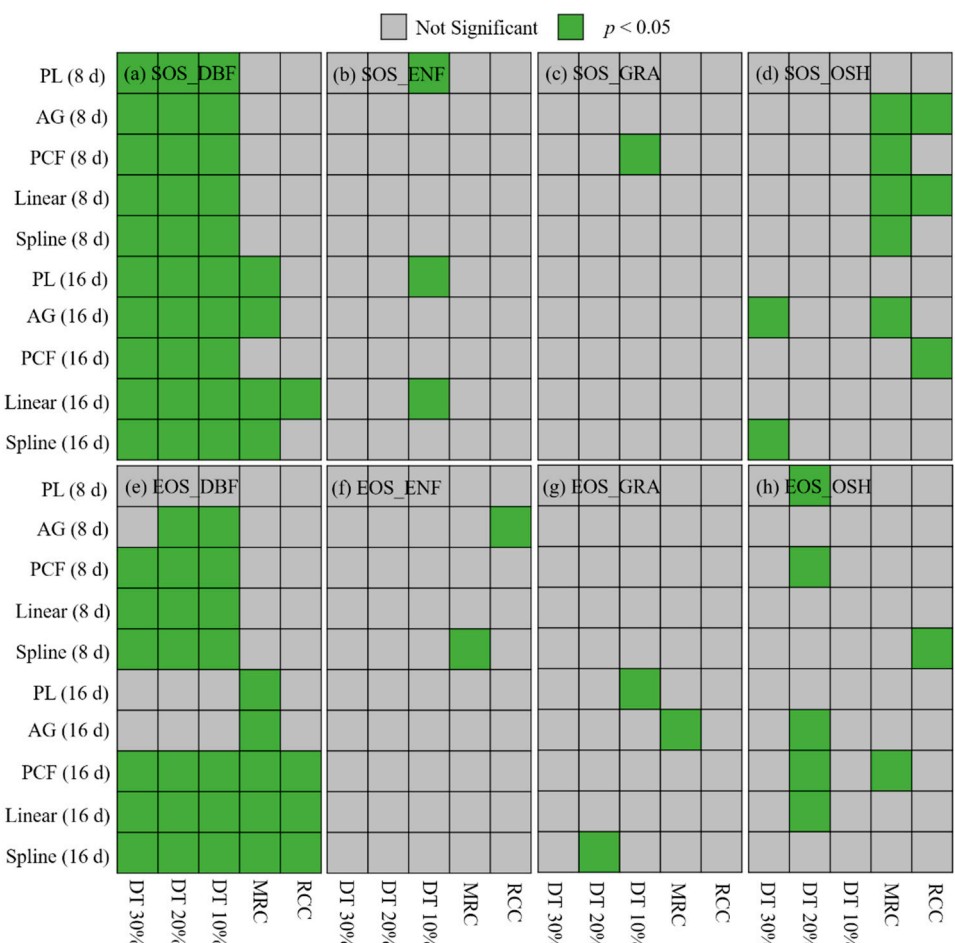

**Figure 5.** Comparisons between phenology trends from daily NDVI data and NDVI composites based on combinations of different time interpolation methods and extraction methods. (**a**) SOS trends comparisons in DBF, (**b**) SOS trends comparisons in ENF, (**c**) SOS trends comparisons in GRA, (**d**) SOS trends comparisons in OSH, (**e**) EOS trends comparisons in DBF, (**f**) EOS trends comparisons in ENF, (**g**) EOS trends comparisons in GRA, and (**h**) EOS trends comparisons in OSH. SOS and EOS are the start of growing season and the end of growing season; DBF, ENF, GRA, and OSH are deciduous broadleaf forest, evergreen needleleaf forest, grassland, and open shrubland, respectively; PL, AG, PCF, Linear, and Spline are piecewise logistic function fitting, asymmetric Gaussian function fitting, polynomial curve fitting, linear interpolation, and cubic spline interpolation, respectively; DT, MRC, and RCC are dynamic threshold, maximum rate of change, and change rate of curvature, respectively; grey boxes indicate that there are no significant differences ($p > 0.05$) between phenology trends from NDVI composites and daily NDVI data; green boxes indicate there are significant differences ($p < 0.05$) between phenology trends from NDVI composites and daily NDVI data.

### 3.4. Comparisons between Trends from the 8-Day and the 16-Day NDVI Composite Data

There were significant differences between phenology trends from the 8-day and the 16-day NDVI composite data (Figure 6) only in DBF and GRA. In DBF, significant differences occurred for all interpolation methods except PCF (Figure 6a). For PL, AG, Linear, and Spline, the mean SOS trends from the 8-day and 16-day NDVI composite data were 0.12 d/year and 0.00 d/year; 0.13 d/year and 0.01 d/year; 0.17 d/year and −0.24 d/year; and 0.22d /year and −0.23 d/year, respectively. There were also significant differences between the mean EOS trends from the 8-day and 16-day NDVI composite data among Linear and Spline, which were 0.07 d/year and −0.13 d/year (Linear), 0.11 d/year, and −0.14 d/year (Spline), respectively (Figure 6e). In GRA, the differences between the mean SOS trend from the 8-day and the 16-day NDVI composite data were significant among Linear and Spline, which were 0.14 d/year and −0.18 d/year (Linear), 0.11 d/year, and −0.20 d/year (Spline), respectively (Figure 6c). In addition, for Spline, the mean EOS trend from 8-day NDVI composite data was −0.09 d/year, which was significantly different from the 16-day mean EOS trend (0.19 d/year) (Figure 6g).

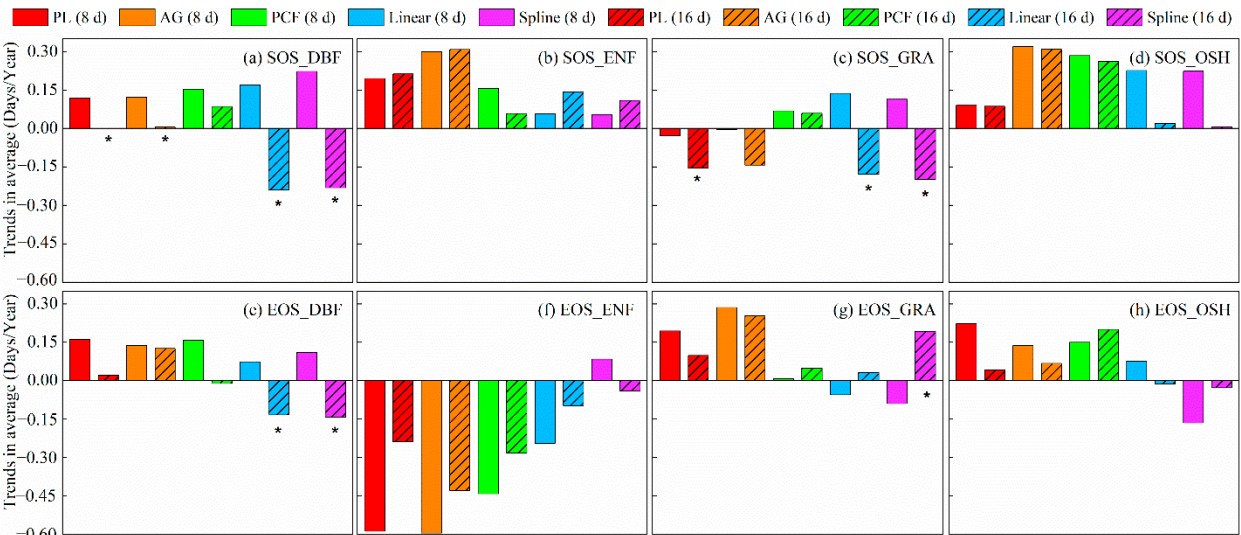

**Figure 6.** Comparisons between mean phenology trends from the 8-day and 16-day NDVI composite data among different interpolation methods. (**a**) SOS trends comparisons in DBF, (**b**) SOS trends comparisons in ENF, (**c**) SOS trends comparisons in GRA, (**d**) SOS trends comparisons in OSH, (**e**) EOS trends comparisons in DBF, (**f**) EOS trends comparisons in ENF, (**g**) EOS trends comparisons in GRA, and (**h**) EOS trends comparisons in OSH. SOS and EOS are the start of growing season and the end of growing season; DBF, ENF, GRA, and OSH are deciduous broadleaf forest, evergreen needleleaf forest, grassland, and open shrubland, respectively; PL, AG, PCF, Linear, and Spline are piecewise logistic function fitting, asymmetric Gaussian function fitting, polynomial curve fitting, linear interpolation, and cubic spline interpolation, respectively; * below the 16-day NDVI composite data indicates there is significant difference ($p < 0.05$) between the mean phenology trends from the 8-day and from the 16-day NDVI composite data.

## 4. Discussion

### 4.1. Effects of Time Interpolation on Trend Estimation among Different Interpolation Methods

Even though differences between the mean trends estimated from NDVI composites and from the reference (daily) data were insignificant, the discrepancies caused by time interpolation could not be ignored. The mean SOS trends based on Linear and Spline from 16-day NDVI composite data were slightly advanced while the mean trend based on daily data was delayed. The mean EOS trends based on Linear from the 8-day NDVI composite data, along with the Linear and Spline from 16-day NDVI composite data, all showed the advancing rates, which were inconsistent with the mean EOS trend based on daily data (showing the delaying rate). Therefore, it might be incomprehensive to

evaluate the effects of multiple interpolation methods only by analyzing the mean trends of all sites. We further calculated the root-mean-square error (RMSE) between the trend estimated from the reference (daily) data and the trend from each interpolation method (Figures S1 and S2). The RMSE values of SOS and EOS trends based on all interpolation methods ranged from 0.35–0.52 d/year and 0.39–0.47 d/year (Figures S1 and S2), which were overall similar between each method, and the piecewise logistic function fitting (8 d) performed slightly better with the lowest RMSE values. For each interpolation method, the ratio of sites for which its absolute values of trends were lower than the corresponding RMSE value ranged from 56 to 77% for SOS trends and 58–71% for EOS trends (Table S2), which implied that the process of time interpolation on NDVI composites might even change the trend direction over half of all sites. RMSE is sensitive to outliers; thus, our calculation results might overestimate the effects of time interpolation on trend estimation, but the uncertainties caused by time interpolation should be considered.

### 4.2. Effects of Time Interpolation on Trend Estimation among Different Vegetation Types

For vegetation types with apparent seasonal changes such as DBF, almost all the time interpolation methods had significant effects on phenology trend estimation. However, for vegetation types with weaker seasonal changes such as ENF, time interpolation methods had almost no significant effects on trend estimation. Figures 7 and 8 showed an example of phenology extraction results and trends from the daily and 8-day NDVI composite data in DBF and ENF, respectively. During 90–150 in Julian day, changes of daily NDVI values and 8-day NDVI composite values in DBF ranged from 0.33 to 0.28 (Figure 7c), while it only ranged from 0.11 to 0.13 in ENF (Figure 7f). Meanwhile, during 270–330 in Julian day, changes of daily NDVI values and 8-day NDVI composite values in DBF ranged from 0.23 to 0.29 (Figure 8c), but it ranged from 0.03 to 0.07 in ENF (Figure 8f). The 8-day NDVI composite data values changed fast in DBF especially during greening and senescence stages (Figure 7b), making it hard for time interpolation to capture the detailed NDVI changes of each Julian day, which increased errors in the extraction of SOS (EOS) annually and in trend estimation (Figure 7a). Compared with DBF, values of NDVI composites changed slightly in ENF (Figure 8b), which resulted in a higher fidelity after time interpolation compared with daily NDVI data. Therefore, the annual extraction results of SOS (EOS) and their trends had higher accuracies (Figure 8a). The same pattern of results was found in our analysis of the 16-day NDVI composite data. We suggest that remote sensing data of daily temporal resolution should be used for estimating phenology trends in vegetation types especially with apparent seasonal changes. For vegetation with weaker seasonal changes, using NDVI composites (i.e., 8-day or 16-day) would have weaker effects on trend estimation.

### 4.3. Effects of Time Interpolation on Trend Estimation among Different Combinations of Time Interpolation Methods and Phenology Extraction Methods

The selection of phenology extraction methods should be fully considered based on study areas, vegetation types, satellite products, and interpolation methods. For vegetation types with apparent seasonal changes such as DBF, even though most time interpolation methods had significant effects on phenology trend estimation, the phenology trends from few specific combinations (i.e., polynomial curve function fitting and maximum rate of change based on the 16-day NDVI composite data in SOS (Figure 5a), asymmetric Gaussian function fitting, and dynamic threshold 30% based on the 8-day NDVI composite data in EOS (Figure 5e)) still showed no significant differences compared with the trends from daily NDVI data. In addition, for vegetation types with weaker seasonal changes such as ENF, there still existed phenology trends from specific combinations that had significant differences with trends from the daily NDVI data. Previous studies also indicated that different combinations could result in different accuracies of trend estimation [33,75,76]. According to our results, the maximum rate of change and the change rate of curvature method could be used for estimating phenology trends of DBF based on 8-day composite NDVI data, while a dynamic threshold of 20% and 30% had a better performance on

phenology trend estimation for ENF. The dynamic threshold of 30% and the change rate of curvature method were suitable for GRA, and a dynamic threshold of 10% had a better result in OSH trend estimation.

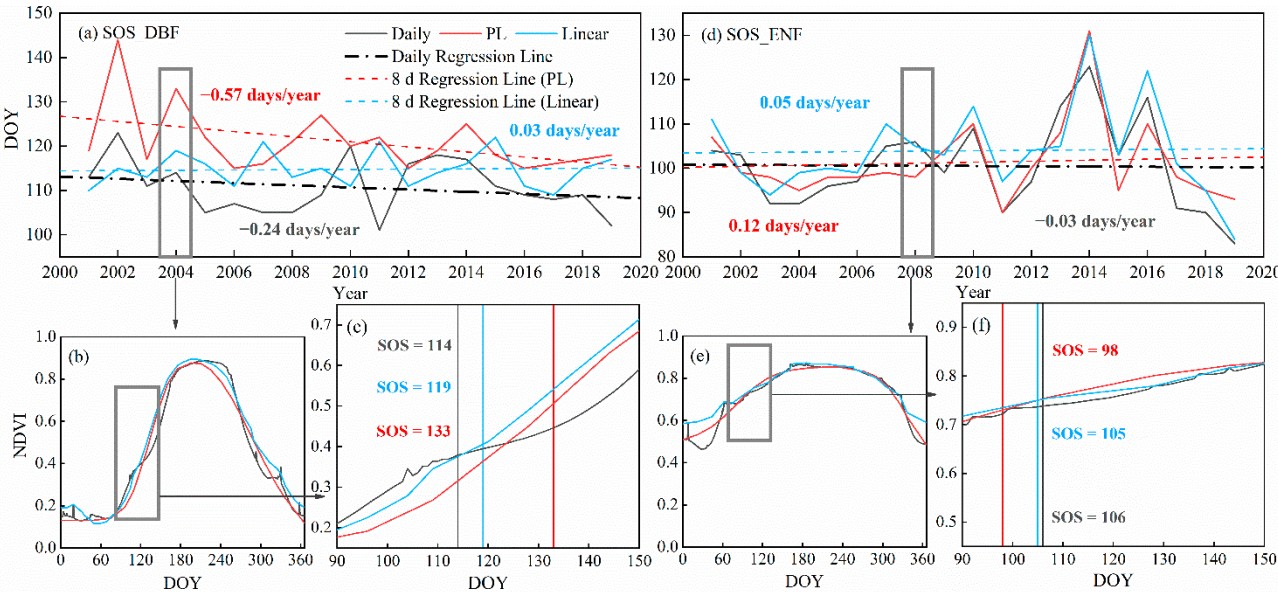

**Figure 7.** Extraction results and trends of the start of growing season (SOS) from the daily and the 8-day NDVI composite data in deciduous broadleaf forest (DBF) and evergreen needleleaf forest (ENF). (**a**) SOS trends of DBF from 2001 to 2019, (**b**) SOS trends of DBF in 2004, (**c**) SOS estimation results of DBF in 2004, (**d**) SOS trends of ENF from 2001 to 2019, (**e**) SOS trends of ENF in 2008, and (**f**) SOS estimation results of ENF in 2008. Piecewise logistic function fitting and linear interpolation are chosen as interpolation method examples; the dynamic threshold 30% is chosen as the extraction method.

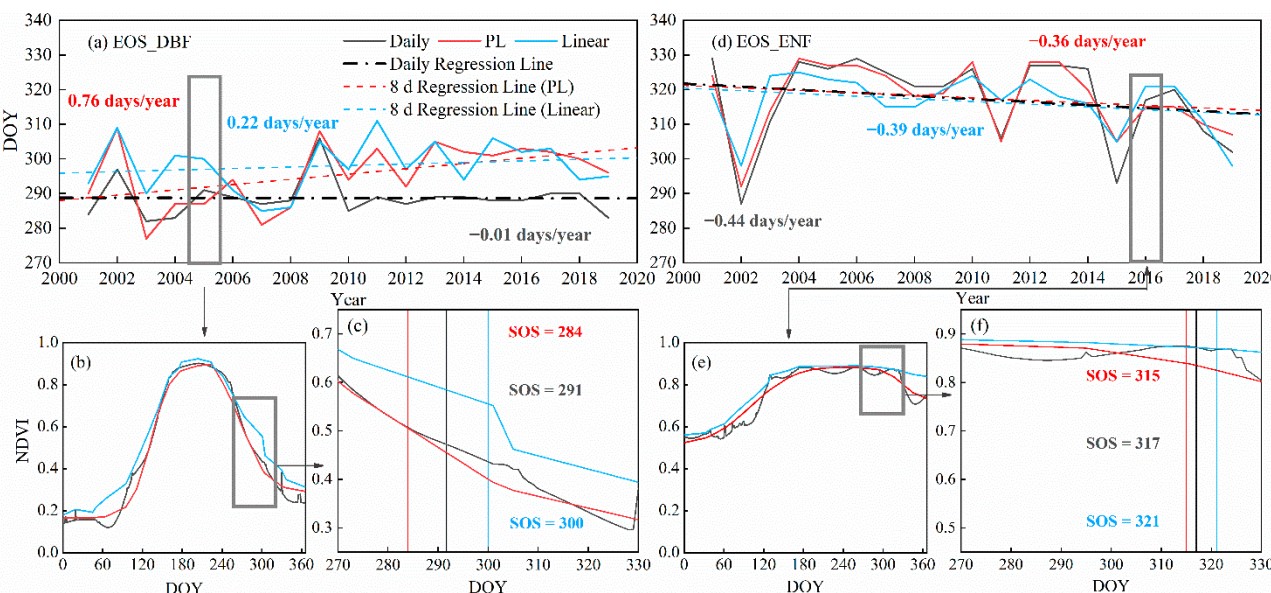

**Figure 8.** Extraction results and trends of the end of growing season (EOS) from the daily and 8-day NDVI composite data in deciduous broadleaf forest (DBF) and evergreen needleleaf forest (ENF). (**a**) EOS trends of DBF from 2001 to 2019, (**b**) EOS trends of DBF in 2005, (**c**) EOS estimation results of DBF in 2005, (**d**) EOS trends of ENF from 2001 to 2019, (**e**) EOS trends of ENF in 2016, and (**f**) EOS estimation results of ENF in 2016. Piecewise logistic function fitting and linear interpolation are chosen as interpolation method examples; a dynamic threshold of 30% is chosen as the extraction method.

### 4.4. Effects of Time Interpolation on Trend Estimation among Data with Different Temporal Resolutions

There were no significant differences between trends derived from the 8-day and 16-day NDVI composite data in ENF and OSH, and significant differences existed in DBF and GRA only among a few interpolation methods. Zhu et al. [8] used asymmetric Gaussian function fitting and piecewise logistic function fitting for estimating SOS trends of Tibetan Plateau alpine meadow from 2000 to 2015, and they found that the trends estimated from the fine temporal resolution (8-day) NDVI data and the coarse temporal resolution (16-day) NDVI data (MOD13A2) had no significant differences, which were in agreement with our research results (Figure 6c,g). Kross et al. [60] also observed that shifts in SOS were not sensitive to temporal resolution (4–28 days) among Canadian deciduous broadleaf sites. According to Figures S1 and S2, the RMSE values of trend estimated from the reference (daily) data and from 8-day NDVI composites were overall lower than those from 16-day NDVI composites. For DBF and GRA, especially among interpolation methods that caused significant differences between trends from the 8-day and 16-day NDVI composite data, we suggest NDVI composites of lower temporal resolution (8-day) for trend estimation when daily-scale datasets were not available. For ENF and OSH, there were no significant differences between trends estimated from the 8-day and 16-day NDVI composite data, which implied fewer discrepancies when applying coarse temporal resolution (16-day) data especially based on the AG and PCF method (Figure 6b,f,d,h) in which the discrepancies were relatively low. However, we only compared the mean trends between the 8-day and 16-day NDVI composite data among all sites; thus, our general conclusions may not apply for every site accurately. The selection of NDVI composites should still be fully considered based on the specific research area, vegetation type, and the data preprocessing method.

### 4.5. Limitations

We observed the delaying SOS mean trends and advancing EOS mean trends among daily-scale NDVI data and NDVI composites, which indicated opposite phenology trends (advancing SOS trends and delaying EOS trends) compared with former research [6,50,52,77,78]. We provide three explanations. First, the results demonstrated above (Figures 3 and 4) were the mean values of SOS and EOS trends, which cannot fully represent the trend of each site with different geographical locations and heterogeneous landscapes. Second, the opposite trends of phenology were also reported by various authors at continental scales over the northern high latitudes due to differences in data sources and scales [8,38], winter or spring warming [73,79], and the mixed impacts of increased spring–fall temperature and fall precipitation [80]. Finally, satellite-based phenological metrics may mainly reflect the spring phenology of early-unfolding (flowering) plant species, indicating that satellite-based phenology trends may follow the trends of ground-measured early plants. Fu et al. [3] found that most of these species showed a delayed trend in spring through the species-specific trend analysis, which confirmed that the delay of the SOS trends monitored by the satellite datasets truly exists. The uncertainties of the opposite phenology trends and their environmental/ecological consequences among different biome zones, study period, and remote sensing sensors still require deeper investigations.

In order to eliminate noise in NDVI time-series curves, we reconstructed the daily NDVI data by using the Savitzky–Golay filter as reference data. However, it still cannot completely simulate NDVI curves in a real natural state, which may cause uncertainties in trend estimation. In addition, similar studies replicated at additional locations, among various satellite products and vegetation types, are also needed for more comprehensive and reliable evaluation on the effects of time interpolation on phenology trend estimation. Due to the mismatch in observation scale (plant scale and pixel scale) and content (the definition of phenological events), we did not use the ground observations as reference data, but comparative studies between using remote-sensing tools and using high-accuracy ground-based measurements still constitute a common and direct method for assessing remote sensing approaches in predicting phenological events. In order to validate vegeta-

tion phenology products properly, ground observations from individual species, canopy cameras, or flux towers should be upscaled temporally and spatially for matching satellite pixels over various ecosystems and geographical regions.

## 5. Conclusions

In this study, we used MODIS MCD43A4 daily surface reflectance data to construct a daily NDVI time-series dataset as the reference data and then generated an 8-day and a 16-day NDVI composite dataset among 120 sites in the mid-high latitudes of the northern hemisphere during 2001–2019. The NDVI composites were used to comprehensively investigate the effects of time interpolation on trend estimation among (1) five time-interpolation methods; (2) four vegetation types; and (3) the combinations of five time-interpolation methods and three extraction methods. We also analyzed the differences of trends estimated between the 8-day and 16-day dataset.

Four main conclusions were drawn from our study. First, none of the interpolation methods had significant effects on trend estimation over all sites, but the discrepancies between trends estimated from NDVI daily data and from NDVI composites could not be ignored. For each interpolation method, the RMSE value of multi-day scale trends was higher than the absolute values of these trends among most of the sites (56–77% of all sites for SOS trends and 58–71% of all sites for EOS trends). Even the effects were insignificant, the process of time interpolation might still change trend direction compared with the trend from the NDVI daily data. Second, time interpolation had significant effects on phenology trend estimation among vegetation types with apparent seasonal changes, but had almost no significant effects among vegetation types with weak seasonal changes. In order to minimize estimation bias, we strongly suggest remote sensing datasets with a daily or high temporal resolution to be applied for estimating phenology trends in vegetation types especially sensitive to season changes. Third, the selection of extraction methods should be fully considered. Trends estimated based on the same interpolation method but different extraction methods were not consistent in showing significant (insignificant) differences with the trend estimated from the daily data, implying that the selection of extraction methods also affected trend estimation. The maximum rate of change and the change rate of curvature method could be used in deciduous broadleaf forest based on 8-day composite NDVI data, while the dynamic threshold of 20% and 30% had better performances for evergreen needleleaf forest. The dynamic threshold of 30% and the change rate of curvature were suitable for grassland, and the dynamic threshold of 10% had a better result in open shrubland. Lastly, for deciduous broadleaf forest and grassland, especially among interpolation methods that caused significant differences between trends from the 8-day and 16-day NDVI composite data, we suggest NDVI composites with a lower temporal resolution (8-day) for trend estimation when daily-scale datasets were not available. For evergreen needleleaf forest and open shrubland, there were fewer discrepancies between trends from 8-day and 16-day NDVI composite data, which implied the availability of using a coarse temporal resolution (16-day), especially based on the asymmetric Gaussian function and the polynomial curve function. In order to further enhance the comprehensive evaluation about the effects of time interpolation on phenology trend estimation, future studies should be carried out at additional locations and among various satellite products and vegetation types.

**Supplementary Materials:** The following are available online at https://www.mdpi.com/article/10.3390/rs13245018/s1, Table S1: *p* values of experimental results in Kolmogorov–Smirnov (K-S) test, Figure S1: Comparisons of the start of growing season (SOS) trends estimated from daily data and from each interpolation method, Figure S2: Comparisons of the end of growing season (EOS) trends estimated from the daily data and from each interpolation method, Table S2: The ratio of sites for which its absolute values of trends were lower than the corresponding root-mean-square error (RMSE) value among each interpolation method.

**Author Contributions:** X.L.: conceptualization, initial analysis, and writing—original draft preparation. W.Z.: conceptualization, funding acquisition, data curation, and writing—review and editing. Z.X.: data curation and writing—review and editing. P.Z.: data curation and writing—review and editing. X.H.: writing—review and editing. L.S.: writing—review and editing. Z.D.: writing—review and editing. All authors have read and agreed to the published version of the manuscript.

**Funding:** This study was funded by the National Natural Science Foundation of China (No. 41771047) and the National Key Research and Development Program of China (Grant No. 2020YFA0608504).

**Conflicts of Interest:** The authors declare that they have no known competing financial interests or personal relationships that could have appeared to influence the work reported in this paper.

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
