# Peer review of "Assessing the Effects of Time Interpolation of NDVI Composites on Phenology Trend Estimation"

_remotesensing, doi:10.3390/rs13245018_

Round 1

Reviewer 1 Report

Comments to the author:

The manuscript is aiming to investigate the effects of time interpolation on phenology trend estimation among different time interpolation methods, vegetation types and the combinations of multiple interpolation methods and extraction methods. The manuscript also analyzed the differences of trends estimated between the 8-day and the 16-day dataset. The results and discussion are in line with the aims set. The manuscript is well written and with fluent English. However, there still need some minor clarifications or modifications for improving the manuscript.

Introduction
• Most important aspects are included. However, the accuracy of the phenology trend estimation also depends on geographical regions, vegetation types and vegetation indexes, which also need to be mentioned. In addition, more latest literatures on the influence of shifts in vegetation phenology should be included

Data and methods
• In 2.1 Study area and sites, the reason for choosing mid-high latitudes as study area requires explanation.

  • In 2.2 Data and preprocessing, for the missing daily NDVI values caused by data loss, the linear interpolation method was applied by using NDVI values of the same day in the nearest years (before and after) to fill up. Why not directly using NDVI values in the nearest days (before and after) for filling up?

  • In 2.2 Data and preprocessing, the reason for choosing Savitzky-Golay filter to smooth the daily NDVI data needs to be provided. And how will this smoothing method influence the further results?

  • In 2.2 Data and preprocessing, the maximum value composite method was chosen to generate a single-year 8-day and a single-year 16-day composite data. 8-day and 16-day are relatively rough in temporal resolution. It is better to generate a higher temporal resolution composite data (i.e., 4-day).

Discussion
• The discussion part should also compare the effects of time interpolation on phenology trend estimation in the start of the season and end of the season. Shifts in spring and autumn phenology trends perform differently towards the process of time interpolation.

  • The study compared the differences of trends between daily NDVI data and multi-day NDVI data, but did not mention the most suitable interpolation method among different vegetation types or among data with different temporal resolutions. Which method leads to higher accuracies in trend estimation, and why is the method most suitable for trend estimation (need literatures).

Reviewer 2 Report

The study focused on shifts in vegetation phenology, which is important for understanding vegetation responses to climate change. Overall, the study is relatively thorough and worth considering to be published. But further improvements are necessary before it is accepted for publication.

Major comments:

1. The introduction needs substantial work to improve the logic and extend the literature review. For example, the second paragraph started with a sentence about "Satellite remote sensing data at various spatial and temporal resolutions", but the rest of this paragraph was not all about this topic. On a related note, introductions about the effect of temporal resolution on phenology extraction (e.g. Peng et al., 2021) were missing. 

Reference:

Peng, D., Wang, Y., Xian, G., Huete, A.R., Huang, W., Shen, M., Wang, F., Yu, L., Liu, L., Xie, Q. and Liu, L., 2021. Investigation of land surface phenology detections in shrublands using multiple scale satellite data. Remote Sensing of Environment252, p.112133.

2. Language reads OK, but it needs further improvement.

3. The study used MCD43A4 product at 500 m resolution and composited the daily data to generate 8-day and 16-day data. I disagree with the reason in Line 123-127. MOD09A1, MOD13A1, and MCD43A4 are registered with the same georeference and the same spatial resolution. 

4. Given the strong contrast of phenology trends based on SOS/EOS calculated using different data and the strong inter-annual fluctuation of SOS/EOS in Figure 8, it would be nice to see more careful calculations of the phenology trends. For example, besides the method in Section 2.3.3, Theil-Sen approach (as used in Huang et al., 2021) is worth trying to reduce uncertainties caused by outliers.

Reference:

Huang, K., Zhang, Y., Tagesson, T., Brandt, M., Wang, L., Chen, N., Zu, J., Jin, H., Cai, Z., Tong, X. and Cong, N., 2021. The confounding effect of snow cover on assessing spring phenology from space: A new look at trends on the Tibetan Plateau. Science of the Total Environment756, p.144011.

Line comments:

Line 69-70, why?

Line 112: How were the Fluxnet sites selected as representative sites?

Figure 4, 5, 6: How was the significance calculated? The method should be included in Section 2.3.3

Reviewer 3 Report

The authors address a crucial issue in the use of remote sensing data for phenology research: how faithful are approximations using multi-day satellite composite time series to the use of daily observations from the same platforms? It seems odd that this question has not been addressed previously, and that 8- and 16-day (and longer-period) composite vegetation index values are taken on faith to be as accurate as necessary, even though new research (like this paper) is finally showing just how accurate those composite products really are.

I have some questions and comments regarding the text:

line 104: "remained" --> "retained"

Section 2.2, subsection (2): how much of the dataset needed linear interpolation in order to fill periods of data loss?

Section 2.2, subsection (4) and Table 1: for each 8- and 16-day period, how was the observation DOY of the selected VI value assigned to that period? For example, many researchers erroneously use the first, middle, or last day of the period as the DOY to assign to that VI value. In the MODIS composite datasets, the DOY corresponding to the actual selected VI value is also provided, and some researchers ignore this field or don't even know it is available. In your analysis using your composite products, do you use the observation DOY corresponding to the selected VI value in each composite period, or (erroneously) a different DOY corresponding to some other aspect of the composite period?

Section 3.1 indicates that the "mean SOS trend of daily NDVI data is 0.07 d/year" and that the "mean EOS trend of daily NDVI is 0.03 d/year." In Section 3.3, the reported trends in the deciduous broadleaf forest are even stronger in the same directions. This suggests that (1) the SOS is retreating, instead of advancing to earlier in the year as a number of studies have shown is consistent with global warming, and (2) that the length of the growing season (EOS - SOS) is shortening, also at odds with studies showing lengthening of the growing season over much of the globe consistent with warming trends. Please address the discrepancy between your calculated results and the expectations according to other studies and the effects of long-term warming.

In Section 4.1 you say that "The mean SOS trends based on Linear and Spline from 8-day NDVI composite data were slightly advanced while the mean trend based on daily data was delayed." With regard to the trends, what do "advanced" and "delayed" mean? And since your daily dataset is the baseline against which characteristics of the other products are compared, what data were used to show a "delay" in the mean trend based on daily data?

lines 325-326: the r-squared values listed are very low, and may only be statistically significant due to large sample sizes. Please address this by discussing those sample sizes and how such small correlations may still be useful.

lines 330-331: the "RMSE values of SOS and EOS trends" should have units of d/year, correct?

line 351: "suited" --> ??? Perhaps "The same pattern of results was found in our analysis of the 16-day NDVI composite data."

line 363 (title of Section 4.3): "... different combinations" of what? "Interpolation and extraction methods" should be stated there explicitly.

lines 402-403: "plant" and "vegetation" are the same thing. Perhaps you mean to offset "plant scale" and "pixel scale"? Or, using 3x3 windows of a 500-m MODIS product, perhaps you mean to offset "plant scale" and "landscape scale."

This is a very suitable paper and a valuable contribution to the literature on the observation of vegetation phenology from remote sensing platforms. I think some clarifications of your methods and results are all you really need here.
